# Perinatal and maternal factors associated with Autism Spectrum Disorder

**Susanna Edlund**[1], **Nils Haglund**[1], **Carl-Gustaf Bornehag**[2,3], **Chris Gennings**[3], **Hannu Kiviranta**[4], **Alexander Kolevzon**[5,6,7], **Christian Lindh**[8], **Panu Rantakokko**[4], **Abraham Reichenberg**[3,5,7], **Shanna Swan**[3]*, **Karin Källén**[1]

1 Institution of Clinical Sciences, Lund, Department of Obstetrics and Gynecology, unit of reproduction epidemiology, Tornblad Institute, Lund University, Lund, Sweden, 2 Department of Health, Karlstad University, Karlstad, Sweden, 3 Department of Environmental Medicine and Public Health, Icahn School of Medicine at Mount Sinai, New York, New York, United States of America, 4 Department of Health Security, Finnish Institute for Health and Welfare (THL), Kuopio, Finland, 5 Department of Psychiatry, Icahn School of Medicine at Mount Sinai, New York, New York, United States of America, 6 Department of Pediatrics, Icahn School of Medicine at Mount Sinai, New York, New York, United States of America, 7 Seaver Center for Autism Research and Treatment, Icahn School of Medicine at Mount Sinai, New York, New York, United States of America, 8 Division of Occupational and Environmental Medicine, Lund University, Lund, Sweden

* susanna.edlund@med.lu.se

## Abstract

### Background

Autism spectrum disorder (ASD) is a heterogeneous neurodevelopmental condition influenced by both genetic and environmental factors. Prenatal and perinatal exposures have been implicated in ASD etiology, but their influence may vary across clinical subgroups, including subgroups defined by co-occurring intellectual disability (ID).

### Methods

We conducted a population-based case–control study in southern Sweden including all children diagnosed with ASD before the age of 9, and whose mothers were born in Sweden. Diagnoses were confirmed through detailed medical record review, and information on ASD severity, ID status, and familial ASD were collected for subgroup analyses. A total of 996 ASD cases and 9,960 age- and sex-matched controls were identified from a regional perinatal database. Multivariable logistic regression estimated adjusted odds ratios (aORs) and 95% confidence intervals (CIs) for maternal, obstetric, and neonatal factors.

### Results

Higher maternal body mass index (BMI) in early pregnancy was associated with increased likelihood of ASD (adjusted odds ratio [aOR] 1.41–1.76 for overweight and obesity compared with normal weight), with broadly similar associations observed

**Data availability statement:** The data underlying this study contain sensitive personal information and cannot be publicly shared due to GDPR and Swedish data protection regulations. Access may be granted upon reasonable request and after approval by the Regional Ethical Review Board in Lund (Dnr 2015/221), with subsequent amendments approved by the Swedish Ethical Review Authority (Dnr 2020-02348). Requests for data access can be directed to registrator@lu.se.

**Funding:** This study was supported by grant R01ES026904 from the National Institute of Environmental Health Sciences (NIEHS). The study was also supported by the Beatrice and Samuel A. Seaver Foundation (Alexander Kolevzon and Abraham Reichenberg).

**Competing interests:** The authors have declared that no competing interests exist.

across ASD subgroups defined by severity, intellectual disability, and familial ASD. Maternal smoking in early pregnancy (aOR 1.49, 95% CI 1.22–1.82) and both elective (aOR 1.22, 95% CI 1.01–1.48) and emergency cesarean delivery (aOR 1.40, 95% CI 1.15–1.81) were also associated with higher odds of ASD, with generally stronger associations in children without intellectual disability and in those with less severe ASD. Subgroup-specific associations were observed for maternal epilepsy and gestational diabetes, while prematurity showed weaker associations than anticipated and was mainly observed in severe ASD and non-familial cases. Low Apgar scores at 5 minutes showed no consistent association with ASD.

## Conclusions

Multiple maternal and perinatal factors were associated with ASD in this large Swedish cohort. Stratified analyses by ASD severity, ID status, and familial ASD revealed both shared and subgroup-specific association patterns, underscoring the value of considering ASD heterogeneity in studies of neurodevelopmental variation.

## Introduction

Autism Spectrum Disorder (ASD) is a neurodevelopmental condition characterized by challenges in social communication and the presence of restricted, repetitive behaviors, including sensory processing difficulties [1–2]. ASD is relatively common, with a prevalence of approximately 2% in Sweden as of 2023 [3]. This aligns with global estimates ranging from 1% to 3% in developed countries [4–5]. The condition presents with a broad range of symptoms and severity levels, reflecting a spectrum defined by shared behavioral features but varying in underlying etiology [6].

Large-scale epidemiological studies report a consistent male predominance in ASD, with male-to-female ratios ranging from 2:1–5:1 [7–8]. Multiple factors may contribute to this disparity, including interactions between genetic susceptibility and environmental exposures, as well as sex-based differences in hormonal and immune function [9–10]. Females with ASD often exhibit more socially normative behaviors—such as greater social motivation, better masking skills, fewer externalizing behaviors, and less atypical interests—which may obscure their difficulties and contribute to underdiagnosis [8,11,12].

ASD presents with a wide range of symptoms and severity levels, reflecting substantial variability in functional abilities, support needs, and underlying causes. This heterogeneity is thought to be shaped by an interplay of genetic, developmental, and environmental influences, including age-related changes in symptom expression, cognitive functioning, behavioral interventions, and access to support services. Genetic effects vary between individuals and often interact with developmental and environmental factors, shaping how symptoms emerge across the spectrum [6,13–17]. Such complexity hinders efforts to clarify the condition's causes, trajectory, and appropriate interventions. At one end of the spectrum, individuals may

present with subtle symptoms and retain strong functional abilities—for example, those with high-functioning ASD often demonstrate average to above-average intelligence and well-developed language skills. At the other extreme, individuals may experience profound impairments, including significant cognitive challenges and minimal language development, which can greatly impact quality of life, autonomy, and daily functioning [18–19]. Common co-occurring conditions include attention deficit hyperactivity disorder (ADHD), language impairments, and various medical and psychiatric disorders such as epilepsy, sleep disturbances, anxiety, and depression [20–22]. In addition, individuals with ASD have a higher risk of chronic health conditions and premature mortality [22–25].

Intellectual disability (ID)—defined by significant limitations in cognitive functioning and adaptive behavior, typically indicated by an intelligence quotient (IQ) below 70 [26]—is the most frequent co-occurring condition after ADHD [20]. In Sweden, approximately 20–25% of individuals with ASD also meet criteria for ID [27]. Individuals with both ASD and ID often require highly specialized and structured support, and they generally have a poorer prognosis. They often face greater challenges in learning, communication, and social interaction, and are at increased risk for severe self-injurious behaviors [28–31].

Understanding the causes of ASD requires integrated consideration of both genetic and environmental contributions [17]. Although no single cause has been identified, it is widely accepted that ASD results from complex interactions between these factors, which may act independently or synergistically during early development. To date, over 800 genes and several genetic syndromes have been linked to ASD, with heritability estimates ranging from 64% to 91%, underscoring a substantial genetic contribution [32–33]. More severe ASD phenotypes are often associated with rare, highly penetrant mutations or copy number variations affecting key neurodevelopmental genes. These cases frequently co-occur with ID [33–34]. In contrast, milder forms of ASD are thought to result from the combined effects of common genetic variants and environmental exposures, reflecting complex polygenic contributions that lead to more subtle neurodevelopmental differences [35]. Such variability is consistent with the broader heterogeneity of ASD, in which different pathways may underlie distinct symptom profiles and functional outcomes [36].

Given this heterogeneity, we reasoned that association patterns might differ according to clinical and familial characteristics. Severity of core symptoms and the presence of intellectual disability have both been linked to distinct genetic architectures and neurodevelopmental pathways, suggesting partly separate etiologic mechanisms. Familial ASD history, in turn, may reflect inherited liability, which could mask or modify associations with environmental exposures. Accordingly, stratifying analyses by these three dimensions—ASD severity, intellectual disability, and familial ASD—was used as an analytic strategy to explore whether prenatal and perinatal factors operate differently across phenotypically and etiologically distinct subgroups.

In line with a neurodiversity-informed perspective, we use terms such as "risk factor" and "association" in their statistical, non-value-laden sense. Our aim is to clarify how different biological and environmental contexts shape neurodevelopmental diversity within the autism spectrum, rather than to imply that autism should be prevented.

Family history is a well-established and important risk factor for ASD, emphasizing its strong genetic basis. Epidemiological studies consistently show that individuals with a first-degree relative diagnosed with ASD are at significantly increased risk themselves, with sibling recurrence rates estimated between 10% and 20%—well above the general population prevalence of 1–2% [37–38]. Twin studies further support a strong genetic influence, with concordance rates of 60–90% among monozygotic twins, compared to much lower rates in dizygotic twins [32,39]. This genetic liability is also reflected in the broader autism phenotype, in which relatives may show subclinical traits such as subtle social difficulties or rigid behavior without meeting full diagnostic criteria [40]. Familial clustering of ASD highlights the strong heritable basis of the condition, while also suggesting that environmental factors may influence how genetic risks are expressed—a perspective supported by recent large-scale epidemiological research [35]. The implications of familial ASD for interpreting risk factor patterns are considered in detail in the Methods and Discussion sections.

 

While an earlier section discussed how developmental and postnatal environmental factors—such as behavioral interventions and access to support services—can shape ASD's clinical presentation, this section focuses on prenatal and perinatal environmental exposures that may contribute to ASD risk. Numerous prenatal and perinatal risk factors have been linked to ASD, including advanced maternal and paternal age [41–43], maternal parity [44–46], and maternal characteristics such as high or low body mass index (BMI) [47–49], as summarized by recent reviews, including that by Katz et al. [50]. Social determinants like low socioeconomic status [51] and ethnicity [52] have also been implicated. Additional risks include maternal smoking [53–54], maternal physical or psychiatric illness [55–56], and obstetric complications such as cesarean section [57], abnormal birth weight [58–60], prematurity [61–63], and certain medications, particularly antiepileptic drugs [64–65] as well as nutritional deficiencies such as low folic acid or vitamin D [66,67]. Finally, environmental pollutants—such as airborne toxins and endocrine-disrupting chemicals—have been associated with increased ASD risk [68–70]. These diverse risk factors may influence early neurodevelopment through a range of biological mechanisms, including oxidative stress, hormonal imbalance, immune dysregulation, epigenetic modifications, and disrupted neuronal growth and connectivity [71–76]. Disruptions during prenatal and perinatal development—periods of heightened neurodevelopmental sensitivity—may therefore contribute to ASD vulnerability [64,77].

Accordingly, using a large, clinically validated population-based cohort restricted to mothers born in Sweden, this study examines maternal, obstetric, and neonatal factors in relation to ASD severity, intellectual disability, and familial ASD. By focusing on subgroup-specific association patterns, the study aims to refine etiological insight and inform how maternal and perinatal health contexts—many of which are already central to routine obstetric care—intersect with neurodevelopmental diversity.

## Methods

### The Skania Autism Study Participants

This study is part of the Skania Autism Study (SAS), a total population-based ASD cohort (N = 996) with linked biosamples, established to support research on environmental risk factors for ASD, particularly focusing on potential exposure to endocrine-disrupting chemicals.

The study population and ascertainment process are summarized in Fig 1.

Children born between 1997 and 2015 who had received a diagnosis of pervasive developmental disorder (ICD-10 code F84, encompassing F84.0 autistic disorder, F84.5 Asperger's syndrome, and F84.9 pervasive developmental disorder, unspecified) were identified through the Scania Child and Youth Habilitation records. Data were extracted retrospectively from routine clinical documentation via the regional electronic medical record system PMO (Profdoc Medical Office; Region Skåne) [78]. No new clinical assessments were performed for the purposes of this study. The habilitation clinics provide cost-free, voluntary support services for children and adolescents with disabilities, and the majority of children diagnosed with autism in the region are referred to these clinics following diagnosis. Only children who were enrolled in a habilitation clinic before the age of nine years at diagnosis were selected (N = 1,991).

Using personal identification numbers, these children were then linked to the Perinatal Revision South (PRS) quality register, which contains data from all obstetric and neonatal units in the southern healthcare region of Sweden [79]. The PRS includes detailed information on maternal and child characteristics, collected at the first antenatal visit (typically at 9–12 weeks of gestation), including maternal age, parity, duration of involuntary childlessness (based on maternal self-report), use of assisted reproduction, chronic illnesses, and lifestyle factors such as smoking and BMI. Maternal BMI was categorized as underweight (<18.5), normal weight (18.5–24.9), overweight (25.0–29.9), or obese (≥30). Maternal smoking was classified as non-smoking, < 10 cigarettes per day, or ≥10 cigarettes per day.

To reduce confounding due to ethnic heterogeneity in the larger environmental study, we restricted the current analysis to children born to mothers who were themselves born in Sweden. This restriction aimed to improve sample homogeneity

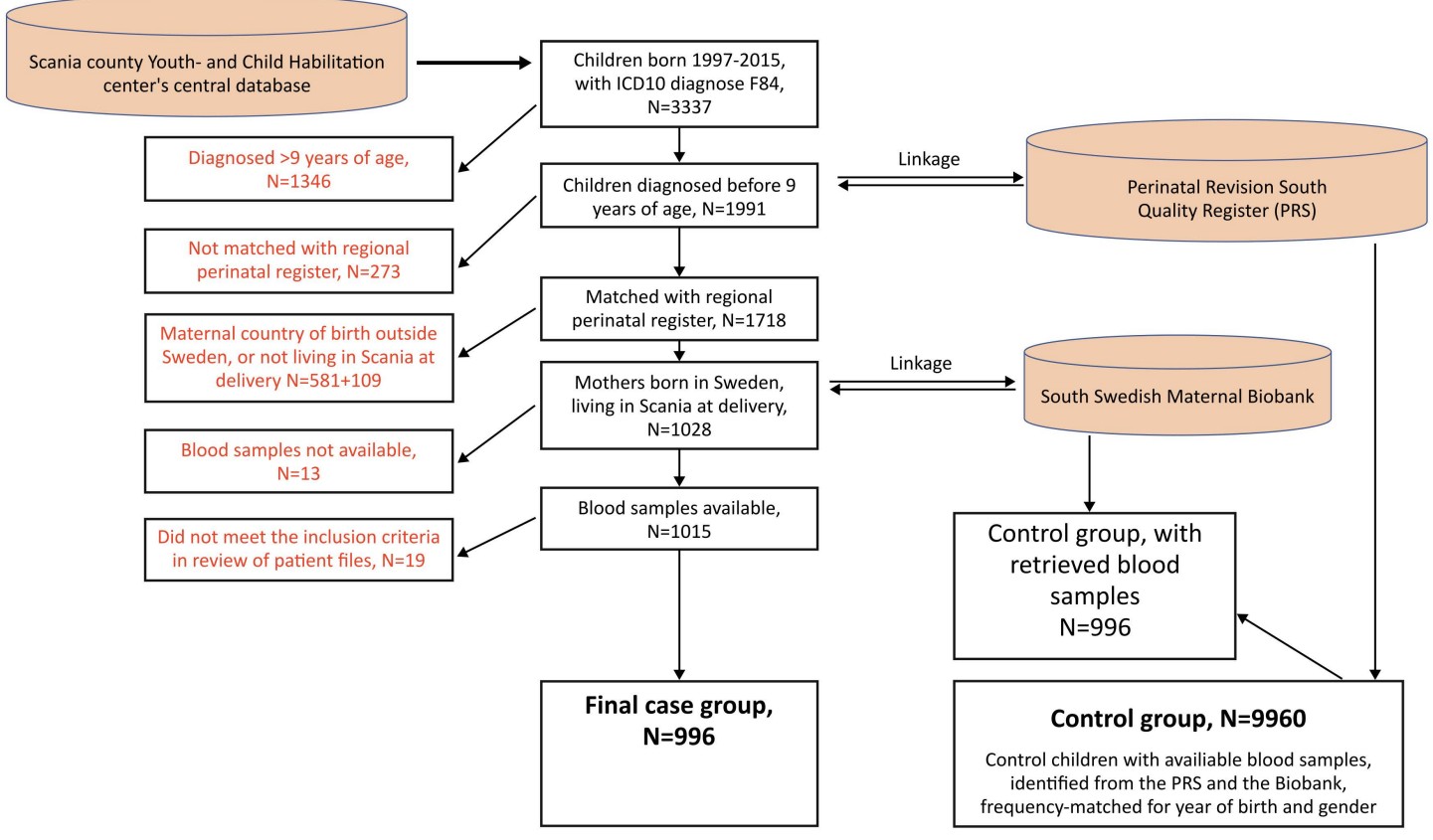

**Fig 1. Flow chart of case and control selection.** The figure illustrates the selection process of ASD cases (n = 996) and matched controls (n = 9,960) from regional clinical, perinatal, and biobank registers in Scania county.

for later analyses of environmental exposures. This yielded 1,028 children, with maternal mid-pregnancy blood samples available for 1,015. The biosamples were routinely collected during early to mid-pregnancy as part of a regional rubella screening program conducted in southern Sweden.

## Diagnosis confirmation

To validate and refine diagnostic accuracy, the medical records of all 1,015 children identified with an ICD-10 F84 diagnosis and available biosamples were reviewed by the first author (S.E.), a licensed clinical psychologist with expertise in neurodevelopmental assessment, using DSM-5 criteria. Children with chromosomal anomalies or confirmed genetic syndromes (e.g., Rett syndrome or Fragile X syndrome) were excluded at this stage, as these cases have a known genetic etiology. Since the primary aim of the study was to investigate environmental risk factors for ASD, such cases were considered potential confounders and therefore excluded from the sample.

Diagnostic evaluation was based on best available clinical and research data. Gold-standard instruments included the Autism Diagnostic Observation Schedule (ADOS) [80–81] and the Autism Diagnostic Interview-Revised (ADI-R) [82]. Additional sources of information included other psychological assessments, behavioral descriptions during clinical visits, and reports from parents, teachers, and other professionals. A confirmed ASD diagnosis was established for 996 children.

ASD severity level, as defined by DSM-5 (Level 1: requiring support; Level 2: requiring substantial support; Level 3: requiring very substantial support), was primarily assessed using documentation from Child and Youth Habilitation Clinics, which provide detailed information on functional abilities in everyday life and required support.

To assess inter-rater reliability, N.H. (co-author), a licensed clinical psychologist with research expertise in autism spectrum disorders, independently reviewed a random sample of 30 patient records. Agreement on severity level was high (Cohen's Kappa = 0.93), with one discordant case resolved by consensus.

ID status was determined from medical records. Most children in the ID group had undergone standardized assessments of IQ and adaptive functioning. In a few cases, children were classified as having suspected ID if they presented with clear cognitive limitations and strong clinical suspicion documented by medical professionals, even if formal assessment was pending at the time of review. Comorbid conditions (e.g., ADHD, cerebral palsy, epilepsy, hearing or language impairment) and familial history of ASD were also recorded.

For the purposes of this study, cases were classified as "familial" if at least one first- or second-degree relative (i.e., parent, sibling, grandparent, aunt, or uncle) had a known ASD diagnosis. Individuals with no such history were classified as "non-familial." Thus, the term *non-familial* refers to the absence of any reported ASD diagnosis among close relatives, rather than missing data. Familial history was determined from anamnestic information in patient records available up to December 2019, including any ASD diagnoses reported in siblings, parents, grandparents, aunts, uncles, or first cousins. Only clearly documented or reliably reported diagnoses (e.g., those confirmed by a medical professional or consistently reported by a parent) were included.

All data were entered and stored in a de-identified REDCap electronic data capture system hosted at Lund University [83].

## Ascertainment of controls

For each child with confirmed ASD, ten controls were randomly selected from the PRS database. Controls were frequency-matched to cases by sex and year of birth and restricted to children born to mothers born in Sweden. Maternal and child characteristics for controls were extracted from the PRS. One randomly selected control from each matched set was scheduled to undergo the same type of maternal blood sample analysis as the ASD cases.

## Statistical analyses

To examine potential risk factors for ASD, we conducted descriptive and regression analyses. Descriptive comparisons of sex, intellectual functioning, reported familial ASD, and age at diagnosis across ASD severity strata were performed using chi-squared tests or Mann–Whitney U tests, as appropriate based on data distribution and variable type. Inter-rater agreement on ASD severity level was assessed using Cohen's Kappa.

Univariable and multivariable logistic regression analyses were conducted to examine associations between maternal, pregnancy, and neonatal factors and ASD diagnosis (overall and by subgroup). Maternal variables included: maternal age (<20 years, 20–34 years [reference], 35–39 years, ≥ 40 years); parity (primiparous vs. multiparous [reference]); smoking during pregnancy (no [reference], yes, unknown); BMI (<18.5, 18.5–24.9 [reference], 25–29.9, ≥ 30, unknown); duration of involuntary childlessness (none [reference], 2–4 years, ≥ 5 years); and assisted reproduction (no [reference], yes). Each risk factor was first assessed in separate univariable models, followed by inclusion in a multivariable model adjusting for all maternal variables. A sensitivity analysis tested for a U-shaped association between ASD and BMI using a second-degree polynomial model. Possible interactions between the covariates were assessed by including interaction terms in the multiple models.

With the scope to investigate a broad range of factors associated with ASD, it was not feasible to apply a strict matching procedure on multiple background variables, as this would have precluded evaluation of the exposures of interest.

Instead, a loose-matching design was applied, in which ten controls were frequency-matched to each case by sex and year of birth only. This approach allowed examination of a wide set of maternal, pregnancy, and neonatal characteristics in relation to ASD. Consequently, unconditional logistic regression with covariate adjustment was used for the main analyses. Methodological studies have shown that for loosely matched case–control designs, unconditional logistic regression provides valid estimates when matching variables and relevant covariates are appropriately included in the models [84].

Pregnancy and delivery complications (entered as binary variables: yes/no) included: preeclampsia; type 1 diabetes; gestational diabetes mellitus (GDM); bleeding during pregnancy; premature rupture of membranes; umbilical cord complications; maternal epilepsy; induction of labor; planned cesarean section (planned CS); emergency cesarean section (emergency CS); vacuum/forceps delivery; and breech presentation. Neonatal risk factors included: birth weight (<2,500 g; 2,500–4,499 g [reference]; ≥ 4,500 g); birth weight relative to gestational age and sex (small [SGA], appropriate [AGA, reference], or large [LGA]); Apgar score at 5 minutes (0–6 vs. 7–10 [reference]); and gestational age (<32 weeks, 32–36 weeks, 37–41 weeks [reference], ≥ 42 weeks).

Included in the multivariable models were obtainable maternal, pregnancy, delivery, and neonatal factors that have previously been associated with ASD or with pregnancy and delivery complications. All regression models were adjusted for the following maternal characteristics: maternal age, parity, smoking during pregnancy, BMI, duration of involuntary childlessness, and use of assisted reproduction. Two-sided p-values <0.05 were considered statistically significant, a conventional threshold commonly used in epidemiological studies to denote evidence against the null hypothesis. Analyses were performed using SPSS version 25 (IBM Corp., Armonk, NY, USA).

### Ethical considerations

Ethical approval for research involving human participants was granted by the Regional Ethical Review Board in Lund (Dnr 2015/221), with subsequent amendments approved by the Swedish Ethical Review Authority (Dnr 2020–02348). The requirement for informed consent was waived.

The authors accessed identifiable patient data solely for the purpose of reviewing medical records, as approved by the relevant ethical authorities. Data were accessed during the approved study period. All data handling, linkage, and storage were conducted in accordance with applicable data protection regulations, including the EU General Data Protection Regulation (GDPR). Data were stored in de-identified form on password-protected media kept in a locked secure cabinet, with access restricted to authorized researchers, and retained in accordance with Swedish research data regulations.

### Patient Involvement

No patients were involved in formulating the research question, selecting outcome measures, or interpreting the results. Study findings will be disseminated through publication in peer-reviewed journals and presentations at scientific conferences.

### Results

The following sections present descriptive and regression analyses examining how ASD risk varies by subgroup and in relation to prenatal and perinatal exposures. Results are presented by cohort composition, ASD characteristics, maternal factors, and neonatal outcomes in that order.

### Sample composition

Table 1 presents the number of ASD cases and controls included in the cohort by matching criteria (year of birth and sex), to reduce potential confounding. The overall male-to-female ratio in the ASD group was 4.6:1, consistent with the pattern described in the Introduction.

**Table 1. Children with ASD and controls by matching criteria (year of birth exactly matched, and sex).**

|  | Children with ASD N=996 | | Controls N=9960 | |
|---|---|---|---|---|
|  | n | (%) | n | (%) |
| **Year of birth** |  |  |  |  |
| 1997-2005 | 355 | (35.6) | 3550 | (35.6) |
| 2006-2010 | 438 | (44.0) | 4380 | (44.0) |
| 2011-2015 | 203 | (20.4) | 2030 | (20.4) |
| **Child sex** |  |  |  |  |
| Males | 820 | (82.3) | 8200 | (82.3) |
| Females | 176 | (17.7) | 1760 | (17.7) |

## ASD severity in relation to age, sex, cognitive function, and familial history

Table 2 shows ASD severity levels by age at confirmed diagnosis, sex, intellectual function, and familial ASD (clearly reported diagnoses in close relatives, see Methods). A significant inverse association was observed between age at confirmed diagnosis and ASD severity, with more severe cases diagnosed at earlier ages. The male-to-female ratio did not differ significantly across severity strata. Among children with ASD and ID, the male-to-female ratio was 4.0:1, compared to 4.9:1 in children without ID. However, this variation across strata was not statistically significant (p-value=0.26; S1 Table).

A significant association was found between cognitive level and ASD severity. ID was present in 16% of children with mild ASD, compared to 45% and 90% in those with moderate and severe ASD, respectively. No significant association was observed between familial ASD and either severity level or presence of ID, a finding that will be further considered in the Discussion (S2 Table).

## Maternal characteristics and ASD risk

Figs 2 and 3 present associations between ASD and maternal characteristics. Fig 2 displays adjusted odds ratios (AORs) for the full ASD group compared to controls, while Fig 3 provides corresponding AORs stratified by ASD subgroup (mild or moderate/severe ASD, with or without ID, and with or without familial ASD). The corresponding numbers and ORs are shown in supplementary S3, S4, and S5 Tables.

**Table 2. Children with ASD, by ASD severity, presence of intellectual disability (ID), and familial history for ASD.**

|  | Total ASD, N=996 | | Mild ASD, N=754 | | Moderate ASD, N=180 | | Severe ASD, N=62 | | p-value (ANOVA) Chi2 |
|---|---|---|---|---|---|---|---|---|---|
|  | n | (%) | n | (%) | n | (%) | n | (%) |  |
| **Age at confirmed diagnosis** |  |  |  |  |  |  |  |  | (<0.001) |
| Years, Mean [SD] | 5.5 | [1.9] | 5.8 | [1.8] | 4.8 | [1.9] | 3.9 | [1.9] |  |
| **Sex** |  |  |  |  |  |  |  |  | 0.101 |
| Males | 819 | (82.2) | 628 | (83.3) | 146 | (81.1) | 45 | (72.6) |  |
| Females | 177 | (17.8) | 126 | (16.7) | 34 | (18.9) | 17 | (27.4) |  |
| **Intellectual function** |  |  |  |  |  |  |  |  | <0.001 |
| Normal | 737 | (74.0) | 632 | (83.8) | 99 | (55.0) | 6 | (9.7) |  |
| ID confirmed/suspected | 259 | (26.0) | 122 | (16.2) | 81 | (45.0) | 56 | (90.3) |  |
| **Familial history** |  |  |  |  |  |  |  |  | 0.852 |
| Non familial ASD | 724 | (72.7) | 550 | (72.9) | 128 | (71.1) | 46 | (74.2) |  |
| Familial ASDFirst or second degree | 272 | (27.3) | 204 | (27.1) | 52 | (28.9) | 16 | (25.8) |  |

After adjustment, maternal age ≥ 40 years was significantly associated with increased ASD risk. This association was evident primarily in children with moderate/severe ASD and those with familial ASD.

Offspring of primiparous women were more likely to have ASD than controls, particularly in the mild ASD and non-ID subgroups.

Maternal smoking during pregnancy was associated with mild ASD across all subgroups—that is, among children with and without ID, and with and without reported familial ASD—and was also associated with familial ASD, with significance retained in the moderate/severe category.

Among the evaluated maternal variables, maternal overweight (BMI 25–29.9), obesity (BMI ≥ 30), and missing BMI data showed the strongest associations with ASD.

A statistically significant association between increasing maternal BMI and ASD risk was observed across nearly all ASD subgroups. Maternal underweight (BMI < 18.5) was associated only with ASD with ID. The possibility of a U-shaped association was explored given prior evidence suggesting both low and high maternal BMI may be linked to atypical neurodevelopment; however, this pattern was not statistically significant in the present cohort (p for quadratic BMI term = 0.880).

No increased risk of ASD was observed among children born to mothers with a history of involuntary childlessness or assisted reproduction.

## Pregnancy and neonatal complications

Pregnancy complications and neonatal characteristics are presented in Figs 4 and 5.

The corresponding tables, showing numbers, crude, and adjusted ORs are presented in supplementary S6, S7, and S8 Tables. An association between "any pregnancy complication" and ASD was observed, but this was attenuated after adjustment for maternal covariates specified in the Methods. Gestational diabetes mellitus (GDM) was the only individual complication initially associated with ASD overall; however, this association was no longer statistically significant after adjusting for BMI. In subgroup analyses, gestational diabetes remained significantly associated with moderate/severe ASD and ASD with ID, even after adjustment for maternal BMI and other covariates (Fig 5).

Maternal epilepsy was associated with moderate/severe ASD and familial ASD. Children with ASD were more frequently delivered by either planned or emergency cesarean section (CS). Emergency CS was associated with all ASD subgroups, whereas elective CS was associated with moderate/severe ASD, and with familial ASD regardless of severity (Fig 5).

Associations between low birth weight and/or very preterm birth (<32 weeks) and ASD were observed but did not remain statistically significant after adjustment for maternal characteristics. For mild ASD, however, significant associations with both very preterm birth and low birth weight (<2,500 g) persisted after adjustment.

No significant association was observed between low Apgar score and ASD.

## Discussion

Autism spectrum disorder (ASD) is a clinically and etiologically heterogeneous condition and identifying consistent environmental risk factors has proven challenging. In this context, the present study aimed to evaluate whether patterns of prenatal and perinatal exposures differed across clinically defined ASD subgroups. Building on prior research that suggests variation in ASD presentation, our goal was to clarify how such variation may relate to early-life exposures. To achieve this aim, we drew on a large, population-based cohort with validated ASD diagnoses and detailed information on prenatal and perinatal exposures, as well as clinical subgroup characteristics based on ASD severity, cognitive function, and familial history. The individually confirmed diagnoses and well-characterized sample allowed for detailed comparisons across subgroups defined by clinical features. Together, these analyses highlight how prenatal and perinatal contexts may contribute to heterogeneity in neurodevelopmental outcomes across the autismspectrum.

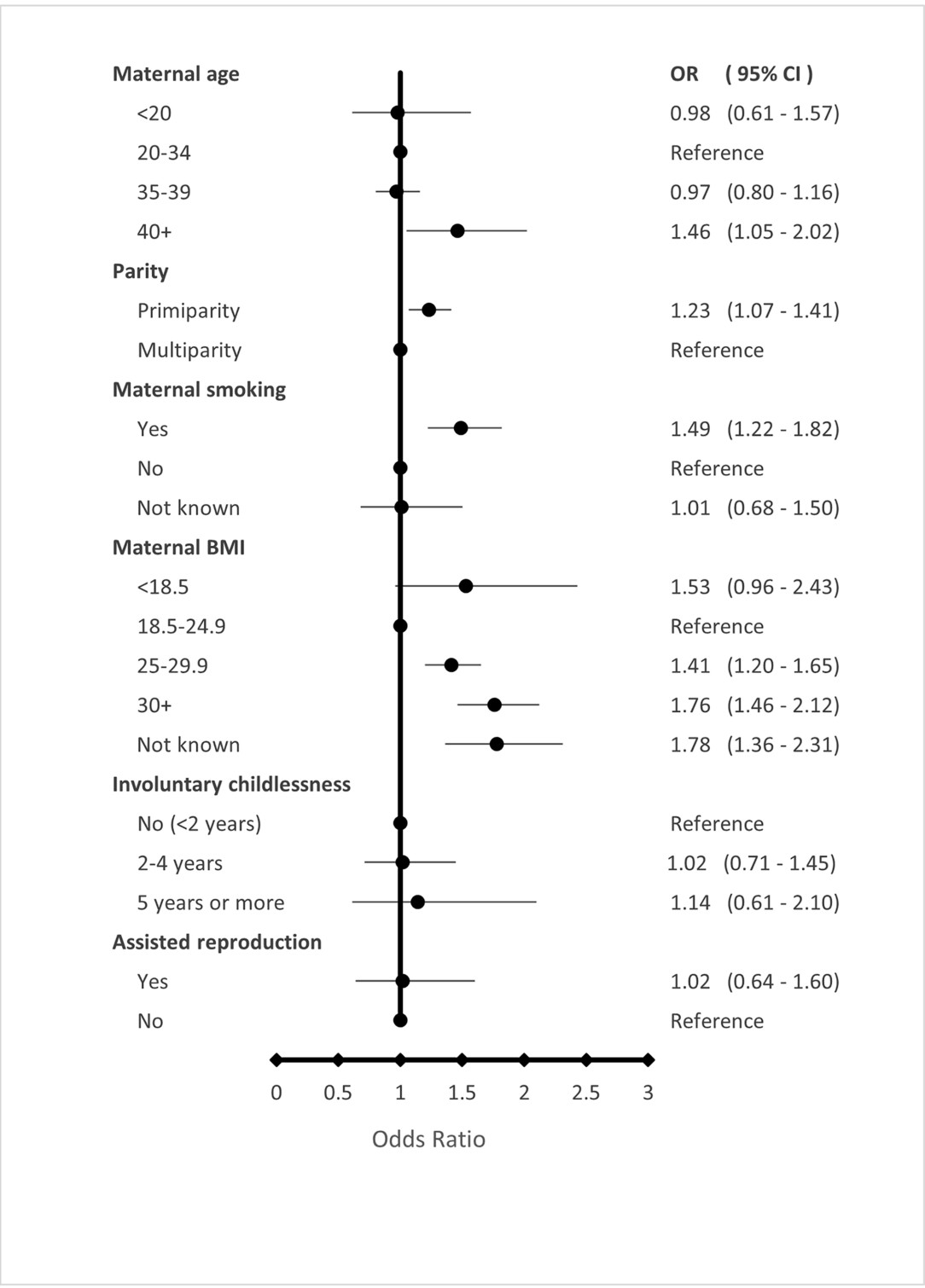

**Fig 2. Associations between ASD and maternal characteristics: adjusted odds ratios (AORs) for the full ASD group compared to controls.**

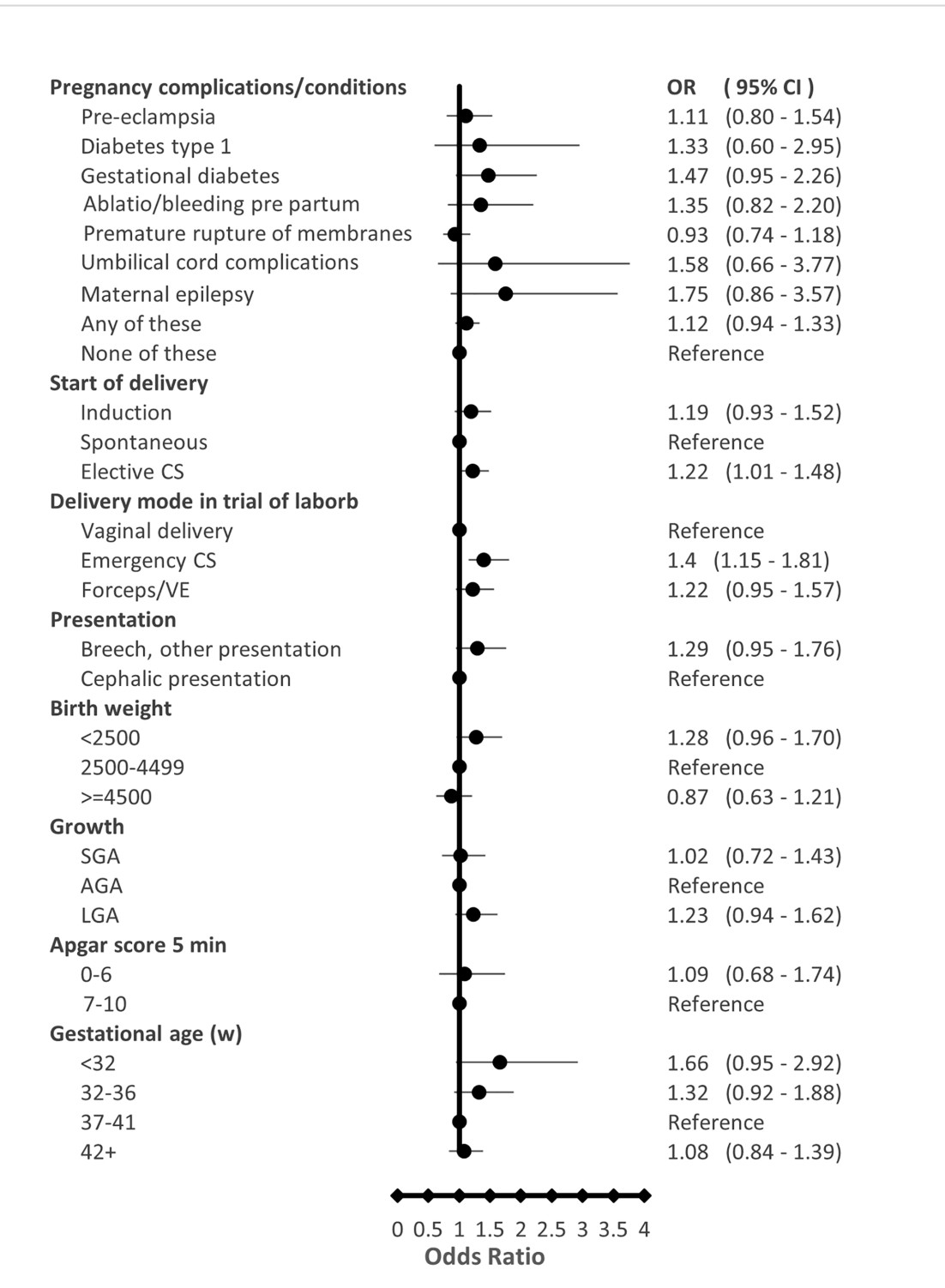

**Fig 3. Associations between ASD and maternal characteristics.** Adjusted odds ratios (AORs) stratified by ASD subgroup (mild or moderate/severe ASD, with och without ID, and with or without familial ASD.

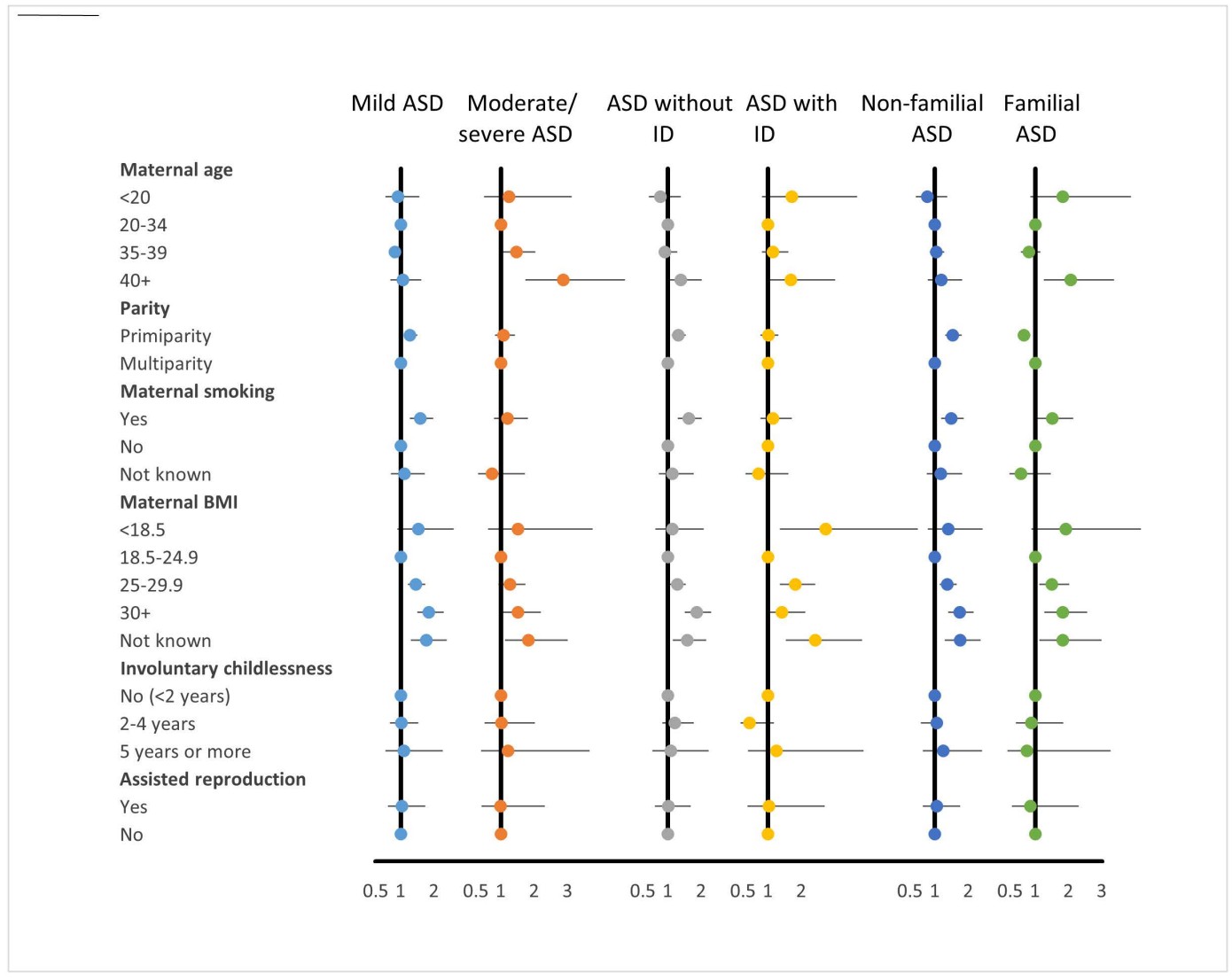

**Fig 4. Pregnancy complications and neonatal characteristics.** Adjusted odds ratios (AORs) for the full ASD group compared to controls.

Maternal smoking, overweight, obesity, elective CS, and emergency CS were factors associated with ASD without any substantial heterogeneity among ASD subgroups. Advanced maternal age was associated with moderate/severe ASD and familial ASD but not in other subgroups. Very preterm birth was associated with mild ASD. One or more ASD subgroups were associated with gestational diabetes, maternal epilepsy and low birth weight [<2500 g].

## Maternal age

Multiple studies have reported an association between advanced maternal age and ASD, with proposed mechanisms including increased risk of chromosomal abnormalities, unstable trinucleotide repeats, and adverse birth outcomes [44,60,85]. Our study found a significant association between advanced maternal age and ASD, despite the exclusion of children with chromosomal abnormalities. This association was evident in both moderate/severe and familial cases and was not explained by pregnancy complications, delivery mode, or prematurity, suggesting that other mechanisms may be involved. One possibility

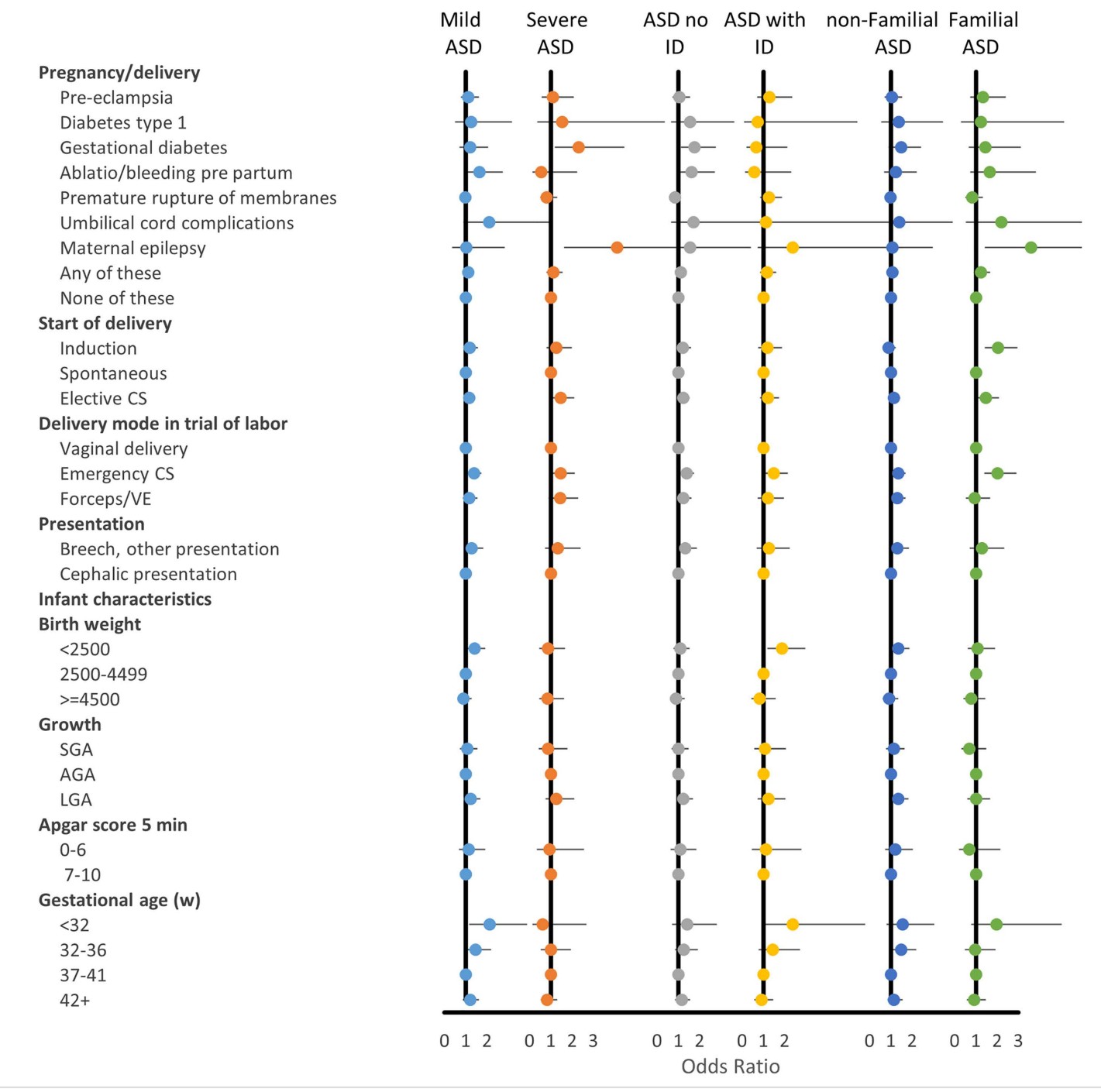

**Fig 5. Pregnancy complications and neonatal characteristics.** Adjusted odds ratios (AORs) stratified by ASD subgroup (mild or moderate/severe ASD, with och without ID, and with or without familial ASD.

is that women with mild ASD traits are more likely to form partnerships later in life. This hypothesis is supported both by our finding that advanced maternal age was more strongly associated with familial ASD, and by previous research showing that maternal age was more strongly associated with ASD cases with reported familial ASD than those without, possibly reflecting delayed childbearing among women with subclinical autistic traits [86]. This interpretation aligns with evidence suggesting that observed associations between advanced parental age and ASD may, at least in part, reflect shared heritable traits and familial mechanisms—such as assortative mating—rather than isolated intrauterine causal mechanisms [87]. In this context, advanced maternal age could serve as a proxy for inherited social, cognitive, or behavioral characteristics that influence both partner selection and timing of reproduction. Accordingly, the observed pattern in the familial subgroup should be viewed as compatible with underlying heritable liability rather than interpreted as a direct effect of maternal age per se.

Advanced paternal age has also been associated with increased ASD risk, likely due to reduced sperm quality, de novo mutations, or epigenetic alterations [88–90]. In Sweden, women are on average 2–3 years younger than their male partners [91], implying that children born to older mothers were also likely to have older fathers. The combined influence of advanced maternal and paternal age has been widely documented, with evidence showing independent contributions from both parents and the highest risk observed when both are older [42,77].

## Primiparity

Primiparity has frequently been linked to an increased risk of several adverse neurodevelopmental outcomes, including ASD [92]. In the present study, primiparity was associated with ASD, particularly among children with mild ASD and those without ID, a pattern that has also been reported previously [45,46,93,94]. One possible explanation is that first pregnancies are more commonly associated with obstetric complications such as preeclampsia, abnormal placentation, and prolonged or difficult labor, which may negatively affect fetal brain development [95–97]. In addition, immunological and hormonal adaptations may differ in primiparous women, with evidence suggesting heightened maternal immune activation and less effective regulatory responses during first pregnancies [98]. Such alterations in immune or stress-related pathways have been implicated in atypical neurodevelopment and increased ASD risk [99–100].

## Maternal smoking

Exposure to tobacco smoke is believed to disrupt fetal brain development through both direct neurotoxic effects and indirect impacts on placental function [101–102]. While some studies suggest that smoking during pregnancy is more strongly linked to other neurodevelopmental or psychiatric conditions than to ASD [103], we found a significant association between maternal smoking and ASD—particularly among children with mild ASD or without co-occurring ID.

These findings are consistent with results from von Ehrenstein and colleagues, who reported a stronger association between heavy smoking and mild ASD than with more severe forms [104]. Notably, their study was not included in a 2015 meta-analysis, which found no measurable association between prenatal smoking and ASD [105]. A more recent meta-analysis from the Environmental Influences on Child Health Outcomes (ECHO) consortium initially found no association either, but after excluding studies with small cell sizes and those including preterm births, a significant association emerged (OR 1.44) [106]. The authors of the ECHO study hypothesized that the third trimester may represent a particularly sensitive window for airborne exposures. Additional evidence supports this focus on subtype and timing. Kalkbrenner et al. reported elevated ASD risk linked to maternal smoking in subtypes characterized by higher cognitive and adaptive functioning [107]. Similarly, a large Swedish registry study by Larsson et al. found maternal smoking was associated with pervasive developmental disorder (PDD), but not with childhood autism or Asperger's syndrome (DSM-IV classification), and that first-trimester exposure alone did not predict increased risk [108].

In our cohort, maternal smoking data were collected during the first trimester. Women who reported smoking were strongly advised to quit, and national statistics from the Swedish National Board of Health and Welfare [109] indicate that

approximately 2.8% of pregnant women smoked at registration in 2023—down from 8.4% before pregnancy—suggesting that nearly half of these women quit in early pregnancy. If later pregnancy smoking is more relevant to ASD risk, and many women ceased smoking after the first trimester, our findings may underestimate the true association. It is important to note that, while our findings point to a potential association between maternal smoking and ASD, whether this reflects a direct effect of tobacco exposure or broader underlying maternal characteristics remains unclear.

**Maternal weight**

The most prominent finding of the current study was the robust and statistically significant association between maternal overweight or obesity and the risk of ASD in offspring. This aligns with findings from Li et al. [110], who reported that maternal obesity was associated with an elevated risk of ASD in a prospective birth cohort, particularly among children with co-occurring intellectual disability. While Li et al. identified a stronger effect in this specific subgroup, our findings indicated a similarly elevated risk across all ASD subtypes, suggesting a more generalized association. In our study, adjustment for other maternal characteristics had only a minimal impact on the observed odds ratios, suggesting that the association between maternal obesity and ASD risk was not substantially confounded by these factors.

Notably, missing BMI data showed one of the strongest associations with ASD. Although speculative, this may reflect underreporting or avoidance of weight measurement among individuals with higher BMI—a phenomenon supported by prior research suggesting that missing maternal BMI is more common among women with overweight or obesity [111–112].

Maternal obesity during pregnancy has been implicated in a range of pathophysiological processes, including systemic inflammation, adiposity-induced inflammatory responses, insulin resistance, altered endocrine signaling, and dysregulation of steroid hormones within the maternal-fetal-placental unit [64,113]. Such disturbances are hypothesized to adversely affect fetal neurodevelopment, which aligns with findings from several studies reporting associations between pre-pregnancy obesity and neurodevelopmental disorders, including ASD [48,49,114]. However, evidence from large-scale meta-analyses suggests that the relationship between maternal overweight and ASD risk may be confounded by familial factors. For instance, a comprehensive meta-analysis found that the association between maternal overweight and ASD was attenuated and became nonsignificant after adjusting for familial confounding, indicating potential genetic or shared environmental influences underlying this link [115]. In contrast, in our study, the elevated risk associated with maternal overweight or obesity persisted within the subgroup of children with familial ASD, and the strength of the association did not differ significantly between familial and non-familial ASD cases.

These findings suggest that maternal BMI may influence ASD risk independently of familial predisposition. Additionally, maternal underweight was associated with increased ASD risk in the subgroup of children with comorbid intellectual disability. This observation aligns with large population-based studies reporting that both extremes of maternal weight—underweight and overweight/obesity—are associated with increased ASD risk in offspring, potentially reflecting a U-shaped relationship between maternal BMI and neurodevelopmental outcomes [116–117]. This interpretation is further supported by evidence of a dose–response pattern, whereby increasing deviation from optimal BMI is associated with heightened ASD risk. Collectively, the evidence supports the cautious conclusion that deviations from optimal maternal weight may contribute to neurodevelopmental vulnerabilities.

**Gestational diabetes mellitus**

In our study, GDM was associated with ASD in crude analysis, but this association did not remain significant after adjusting for maternal BMI. This attenuation suggests that maternal BMI may mediate or confound the relationship between GDM and ASD, potentially explaining the lack of an independent association in our adjusted models. This pattern aligns with previous research suggesting that maternal metabolic factors may confound the relationship between GDM and ASD [118]. Recent meta-analyses have reported a modest but statistically significant association between GDM and ASD,

typically with pooled odds ratios in the range of 1.3 to 1.5 [119–120]. However, these findings are accompanied by substantial heterogeneity, likely stemming from differences in study populations, analytical approaches, and methodological quality, as well as potential publication bias. Notably, Wan et al. [120] found that the association persisted in high-quality case-control studies, suggesting that a link may exist under specific conditions. Our findings are more consistent with large population-based cohort studies, such as that by Xiang et al. [121], in which the association between GDM and ASD was attenuated after adjusting for gestational age, delivery mode, and maternal BMI.

Inconsistencies across studies may further reflect variations in diagnostic criteria, timing and severity of GDM, study design, and the extent to which confounding factors—such as maternal BMI or delivery-related variables—were accounted for.

## Maternal Epilepsy

Epilepsy has been consistently linked to increased risks of neurodevelopmental disorders, including ASD in offspring [122]. In the present study, no significant association was found between maternal epilepsy and ASD in the overall cohort. However, subgroup analyses revealed significant associations in children with familial ASD and those with moderate to severe symptoms. This aligns with prior findings suggesting that the neurodevelopmental impact of epilepsy may be more pronounced in ASD subgroups defined by severity or cognitive impairment [122–123]. Few studies have addressed familial ASD status in the context of maternal epilepsy, highlighting the novelty of our findings. Importantly, the observed associations may not be attributable to epilepsy itself, but rather to related factors such as prenatal exposure to antiepileptic drugs (AEDs).

A key concern is the teratogenic potential of certain AEDs, particularly valproic acid, which is associated with congenital malformations and delayed neurodevelopment [124–126]. Notably, valproate exposure during pregnancy independently increases ASD risk, even after adjusting for maternal epilepsy [122]. Recent large-scale register-based studies further confirm this pattern, showing that prenatal exposure to valproate—but not lamotrigine or topiramate after confounder adjustment—is associated with significantly elevated ASD risk [127,128]. Distinguishing the effects of epilepsy itself from those of AED treatment remains challenging, and further research is warranted to clarify the mechanisms underlying the observed associations.

## Elective cesarean section

Elective CS was more common among children with ASD than controls. In the absence of pregnancy complications requiring planned CS, this association may reflect maternal psychological characteristics. In a large survey, autistic pregnant individuals were about three times more likely to report prenatal anxiety than non-autistic peers (48% vs. 14%) [129]. Moreover, elevated anxiety has been reported in subsequent pregnancies among mothers who already have a child with ASD [130–131]. Such factors may influence preferences or decisions related to elective delivery. Our findings, based on detailed adjustment for potential confounders, strengthen and extend previous evidence of a significant association between elective cesarean delivery and ASD [57,132]. Notably, these and other prior studies often did not clearly differentiate between elective and emergency cesarean deliveries, limiting interpretation of whether distinct mechanisms may be involved. It should be noted that associations between cesarean delivery and ASD may partly reflect the medical indications prompting surgery rather than the procedure itself. This possibility—often referred to as confounding by indication—has been demonstrated in large meta-analysis and population-based cohort studies, including family- and sibling-informed designs, where associations were substantially attenuated after adjustment for obstetric complications and shared familial factors [57,132–134].

## Emergency cesarean section

We also found that children with ASD were significantly more likely to be delivered by emergency CS than controls, suggesting that although no specific pregnancy complications were recorded, deliveries with children with ASD were more

likely to have been exposed to delivery-related distress or complications not captured by recorded diagnoses. Our finding that emergency CS was associated with ASD across all subgroups, while elective CS was linked primarily to more severe and familial ASD presentations, is partly consistent with literature indicating only modest overall risk differences by CS type. Large registry studies have reported similar ASD risk estimates for both elective and emergency CS (adjusted OR ≈ 1.2–1.3), and sibling analyses suggest that such associations may largely reflect familial confounding [133,135,136].

## Apgar score

Despite the assumption that emergency CS might indicate fetal distress, we found no association between low Apgar scores at 5 minutes and ASD. While some large cohort studies have reported modestly elevated ASD risk with low Apgar scores (e.g., RRs 1.4–1.5) [137–138], our null findings align with umbrella reviews suggesting that this association is weak and inconsistent [139]. Such discrepancies in findings may reflect differences in the handling of confounding, measurement of ASD outcomes, timing and thresholds used for Apgar scoring, or variation in cohort characteristics. This highlights the need for cautious interpretation of Apgar score as an independent ASD risk marker.

## Preterm birth

Preterm birth has been consistently associated with an increased risk of ASD in numerous epidemiological studies. Large-scale population-based cohorts have reported a two- to threefold increase in risk among children born preterm, particularly among those born very or extremely preterm [58,63,140]. Additionally, the degree of prematurity has often been reported to correlate with ASD symptom severity, suggesting a dose–response relationship [141]. A recent large Nordic register study found a continuous increase in ASD risk across the full gestational age range, with the lowest risk at 40 weeks, suggesting that even modest reductions in gestational duration may influence neurodevelopment [142].

In contrast, our findings diverged from this established pattern: we found no general association between ASD and prematurity (including both very preterm and moderately to late preterm birth), but did observe an association between very preterm birth and mild ASD. This finding is in line with a study by Joseph et al. [61], who reported that extremely preterm children (born at 23–27 weeks' gestation) had a markedly elevated prevalence of ASD (7%) and that a substantial proportion of these children presented with milder forms of ASD, often without comorbid intellectual disability. These results suggest that very preterm birth may be associated not only with increased ASD risk but also with specific clinical profiles characterized by less severe symptomatology. Taken together with our own findings, this raises the possibility that preterm birth may contribute to a distinct, milder ASD phenotype in a subset of children.

Beyond these associations, few pregnancy complications and perinatal conditions were significantly over-represented in the ASD group after adjustment for confounders. This is consistent with prior population-based studies suggesting that many perinatal factors—such as gestational hypertension, preeclampsia, or low Apgar score—may not independently predict ASD when accounting for maternal characteristics and delivery complications [44,60,143].

## Familial ASD and etiological perspectives

To explore potential heterogeneity in ASD risk factors, we stratified cases by familial ASD history to distinguish between probable high versus low genetic predisposition. The rationale was that strong environmental risk factors may be masked in individuals who would have developed ASD regardless of perinatal exposures. Our results indicated a stronger association between maternal epilepsy and ASD in the familial subgroup, suggesting that epilepsy may be part of the broader genetic spectrum linked to ASD. By contrast, no heterogeneity was observed for gestational diabetes or emergency CS, implying that these may represent independent environmental risk factors. Similarly, maternal smoking and high BMI were associated with ASD in both familial and non-familial cases. The comparable strength of these associations supports their potential role as generalizable, modifiable risk factors across ASD subtypes.

In our sample of confirmed ASD cases, the proportion of children with co-occurring intellectual disability (ID) did not differ significantly between familial and non-familial cases (22.8% vs. 27.2, p = 0.157). This contrasts with prior studies suggesting that familial aggregation is stronger for ASD without ID [144–145]. Notably, while our analysis focused on the prevalence of ID within ASD cases stratified by familial history, studies such as Xie et al. [145] estimated familial recurrence risk in relatives and found stronger genetic clustering for ASD without ID. These methodological differences—case-stratified prevalence versus family-based recurrence estimates—may partly explain the diverging patterns. They also underscore the need for further research on how intellectual disability influences the familial architecture of ASD.

A recent study using the SPARK cohort (Simons Foundation) identified four biologically distinct ASD subtypes with different patterns of genetic variation and developmental disruption [146]. These findings underscore that ASD encompasses a range of etiological pathways and clinical presentations. While the SPARK study emphasizes genetic heterogeneity across subtypes, our findings highlight environmental factors—such as maternal smoking, elevated BMI, and emergency CS—that showed consistent associations across phenotypic subgroups. In contrast to SPARK's genetic classification, our approach used observable clinical characteristics and modifiable perinatal exposures to explore etiological variation. Together, these perspectives underscore the widely proposed importance of integrating genetic and environmental frameworks to advance understanding of ASD etiology [17,37,39,147].

## Novelty and contributions

In this context, the present findings extend previous Scandinavian registry studies on ASD by combining detailed clinical characterization with a focus on maternal and perinatal factors that vary across clinically meaningful subgroups. Unlike large registry-based studies that rely solely on diagnostic codes, the current study used individually confirmed DSM-5 diagnoses, enabling refined subgroup analyses by ASD severity, intellectual disability, and familial ASD.

Our results both confirm and refine previously observed associations, providing a more nuanced understanding of how maternal and perinatal factors operate across ASD subtypes. In line with population-based and sibling-comparison studies by Zhang and colleagues, associations between cesarean delivery and ASD appear largely explained by familial confounding and shared background factors [134]. In contrast, maternal overweight, obesity, and smoking in our analyses showed association patterns that were consistent across ASD subgroups, suggesting additive environmental influences that extend beyond inherited liability. By distinguishing between familial and non-familial ASD, the present analyses complement the work by Xie and colleagues, who reported stronger familial clustering for ASD without intellectual disability [145]. While their study emphasized genetic aggregation, our findings highlight maternal and perinatal factors that may contribute independently of familial background.

The study further contributes new evidence on gestational age and ASD by contextualizing our results in relation to prior Nordic cohort research by Persson and colleagues [142]. Whereas Persson et al. reported a continuous increase in ASD prevalence across the full range of prematurity, our findings suggest that this pattern is primarily evident in milder ASD presentations. This indicates that shortened gestation may be linked to specific neurodevelopmental profiles rather than a uniform increase across the autism spectrum.

Finally, by restricting the cohort to mothers born in Sweden, we reduced heterogeneity related to ethnicity and migration, thereby enhancing internal validity and comparability within the Nordic healthcare context. Together, these features underscore the complementary and novel contribution of the present study relative to existing large-scale register-based research.

While the present study was not designed to investigate underlying mechanisms, the subgroup patterns observed suggest that distinct biological or environmental pathways may contribute to ASD heterogeneity. Future work integrating clinical data with biological markers—such as metabolic, inflammatory, or pharmacological exposures—could help clarify these mechanisms.

## Strengths and limitations

This study's main strengths include its large, population-based sample, clinically validated ASD diagnoses, and linkage to detailed prenatal and perinatal data collected prospectively. The ability to analyze subgroup differences based on severity, intellectual disability, and familial history adds nuance to the findings. Additionally, the Swedish registry system allows for near-complete population coverage, reducing selection bias and strengthening internal validity within the studied population.

Several limitations should also be acknowledged. First, detailed paternal characteristics—such as age, body mass index (BMI), and smoking—were not available, which may have contributed to unmeasured familial or environmental confounding. Second, information on parental socioeconomic status and educational level was lacking, limiting our ability to adjust for social determinants that may correlate with both maternal health characteristics and ASD diagnosis. Third, some key exposures, including maternal smoking, were based on self-report and may therefore be subject to underreporting. Although maternal BMI was measured at antenatal registration, missing BMI values were relatively common and may not have been missing at random, potentially reflecting systematic differences in maternal characteristics.

Fourth, certain pregnancy and neonatal variables—such as gestational weight gain and some obstetric complications—were available only as binary indicators, which may have obscured more nuanced associations. Fifth, although restricting analyses to Swedish-born mothers reduced heterogeneity related to migration and environmental exposures, this design choice may limit generalizability to more diverse populations. As in all observational studies, the regression analyses assume that the specified covariates and functional forms adequately capture the relationships between exposures and ASD. Residual confounding and model misspecification cannot be entirely excluded, particularly given the absence of some potentially relevant covariates. Finally, familial ASD status did not distinguish between maternal and paternal lineage. This limits interpretation of findings observed specifically in the familial subgroup—such as associations with maternal age, elective cesarean section, and maternal epilepsy—and precludes conclusions about whether these patterns reflect parental confounding or gene–environment interactions. Future studies incorporating lineage-specific familial data would allow more refined etiological interpretation.

As in all observational studies, the regression analyses assume that the specified covariates and functional forms adequately capture the relationships between exposures and ASD. Residual confounding and model misspecification cannot be entirely excluded, particularly given the absence of some potentially relevant covariates. Some subgroup strata were relatively small—particularly severe ASD and severe ASD with co-occurring intellectual disability—which resulted in reduced statistical power and wider confidence intervals. Consequently, subgroup-specific estimates should be interpreted with caution, especially when confidence intervals are wide, as limited precision may obscure or exaggerate associations. In addition, the study involved multiple comparisons across numerous exposures and subgroup analyses, which increases the possibility of chance findings. We therefore emphasize interpretation based on effect sizes, confidence intervals, and consistency of patterns across related analyses, rather than on statistical significance alone. Notably, several associations showed consistent direction and magnitude across subgroups, supporting the robustness of the main findings despite limited power in the smallest strata.

Beyond the limitations discussed above, a broader consideration relates to how ASD subgroups are defined and used in research. ASD is a heterogeneous condition, and while subgrouping can help clarify meaningful patterns, classification systems vary widely depending on available data, methodology, and study aims. These subgroup definitions often lack longitudinal validation and may not remain stable over time. Findings should therefore be interpreted as reflecting general trends rather than absolute distinctions, particularly when applied to individual cases.

## Conclusion

This comprehensive, population-based epidemiological study identified several maternal and perinatal factors associated with ASD, consistent with previous research. Notably, associations between maternal overweight/obesity and smoking with ASD were consistent across nearly all ASD subgroups. Surprisingly, prematurity was not a significant risk factor in the

overall cohort—only in the mild ASD subgroup. In general, subgroup analyses revealed smaller differences than anticipated. Future research would benefit from longitudinally designed studies that integrate diverse biological, social, and environmental data. Greater contextual detail—including family background, behavioral profiles, and paternal factors—may help clarify ASD risk mechanisms and improve subgroup classification.

## Supporting information

**S1 Table. Sex by ASD severity, intellectual disability, and familial history for ASD.**
(DOCX)

**S2 Table. Familial and non-familial ASD in relation to presence of intellectual disability among children with ASD.**
(DOCX)

**S3 Table. Children with ASD and controls by matching criteria (year of birth and sex), and maternal characteristics.**
(DOCX)

**S4 Table. Autism severity, presence of intellectual disability, and familial history, respectively, by maternal characteristics.**
(DOCX)

**S5 Table. Autism severity, presence of intellectual disability, and familial history, respectively, by maternal characteristics.**
(DOCX)

**S6 Table. Children with ASD and controls, by presence of pregnancy, delivery complications, and infant characteristics.**
(DOCX)

**S7 Table. Autism Spectrum Disorder (ASD) severity, familial history, and presence of Intellectual Disability (ID), respectively, by pregnancy complications, delivery mode, and infant characteristics.**
(DOCX)

**S8 Table. Autism Spectrum Disorder (ASD) severity, familial history, and presence of Intellectual Disability (ID), respectively, by pregnancy complications, delivery mode, and infant characteristics.**
(DOCX)

## Author contributions

**Conceptualization:** Susanna Edlund, Nils Haglund, Carl-Gustaf Bornehag, Chris Gennings, Alexander Kolevzon, Hannu Kiviranta, Christian Lindh, Panu Rantakokko, Abraham Reichenberg, Shanna Swan, Karin Källén.

**Data curation:** Susanna Edlund, Christian Lindh, Karin Källén.

**Formal analysis:** Susanna Edlund, Karin Källén.

**Funding acquisition:** Nils Haglund, Carl-Gustaf Bornehag, Chris Gennings, Alexander Kolevzon, Hannu Kiviranta, Christian Lindh, Panu Rantakokko, Abraham Reichenberg, Shanna Swan, Karin Källén.

**Investigation:** Susanna Edlund, Chris Gennings, Abraham Reichenberg, Karin Källén.

**Methodology:** Susanna Edlund, Nils Haglund, Chris Gennings, Alexander Kolevzon, Abraham Reichenberg, Shanna Swan, Karin Källén.

**Project administration:** Christian Lindh, Abraham Reichenberg, Karin Källén.

**Supervision:** Abraham Reichenberg, Karin Källén.

**Validation:** Susanna Edlund, Nils Haglund, Alexander Kolevzon, Karin Källén.

**Writing – original draft:** Susanna Edlund, Karin Källén.

**Writing – review & editing:** Nils Haglund, Carl-Gustaf Bornehag, Chris Gennings, Alexander Kolevzon, Hannu Kiviranta, Christian Lindh, Panu Rantakokko, Abraham Reichenberg, Shanna Swan, Karin Källén.

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
