## [Decision Letter · Decision Letter 0]

28 Feb 2025

Dear Dr. Edlund,

Thank you for submitting your manuscript to PLOS ONE. After careful consideration, we feel that it has merit but does not fully meet PLOS ONE’s publication criteria as it currently stands. Therefore, we invite you to submit a revised version of the manuscript that addresses the points raised during the review process.

We look forward to receiving your revised manuscript.

Kind regards,

Mu-Hong Chen, M.D., Ph.D.

Academic Editor

PLOS ONE

Journal Requirements:

2. Thank you for stating the following financial disclosure: This study was supported by grant R01ES026904 from the National Institute of Environmental Health Sciences (NIEHS). The study was also supported by The Beatrice and Samuel L Seaver foundation (AR and AK).

3. In the online submission form, you indicated that the register data are not publicly available due to privacy protection, including General Data Protection Regulations (GDPR). Access to Swedish register resoruces is only granted after ethical review by appropriate authorities. Requests should be directed to registrator@lu.se.

Reviewers' comments:

Reviewer's Responses to Questions

**Comments to the Author**

1. Is the manuscript technically sound, and do the data support the conclusions?

Reviewer #1: Partly

Reviewer #2: Yes

2. Has the statistical analysis been performed appropriately and rigorously?

Reviewer #1: No

Reviewer #2: Yes

3. Have the authors made all data underlying the findings in their manuscript fully available?

Reviewer #1: Yes

Reviewer #2: No

4. Is the manuscript presented in an intelligible fashion and written in standard English?

Reviewer #1: No

Reviewer #2: Yes

Reviewer #1: Overview:

This paper utilises comprehensive population-based data with validated diagnostic variables in an attempt to provide an overview of maternal and perinatal risk factors for ASD. It is commendable in its thoroughness in identifying rpotential risk factors and carrying out an extensive sets of analyses.

However, it is the reviewer’s opinion that the manuscript in its current form suffers from a number of significant shortcomings in terms of methodological problems, unclear or erroneous presentation of currently available evidence, methods, and results. Large-scale revising including redo of the main analyses with appropriate statistical models are needed before it may be deemed publishable. Comments are categorsied by sections as below:

1. Abstract needs to be reformulated. There are texts that offered little information (for example, ‘following thorough review of medical records’), or are confusing out of context (for example, the mention of cases being ‘classified based on level of intellectual disability’ without explaining the relevance of intellectial disability first). Findings other than those concerning maternal weight (for example, regarding maternal smoking and delivery mode) were not described.

2. Introduction: overall, this section is too short and fails to cover crucial grounds. It should starts with a description of the disease entity (it's epidemiology, clinical presentation, burden etc) and goes on to discuss available evidence relevant to the research question(s) of this study. The section usually concludes with an exposition of the aims and objectives of the study.

2.a. The discussion on available evidence on maternal and perinatal risk factors for ASD, the grand theme of the current manuscript, is very scarce (a mere paragraph of one sentence). In-depth description of strength of evidence, important latest available studies (large-scale longitudinal studies or systematic reviews such as Hisle-Gorman et al. 2018), gaps in the current evidence base relevant to your paper should be described.

2.b. The discussion on ASD+ID is necessary as ASD with or without ID is analysed separately later on, but the importance of doing so is not properly established here. Which ‘worse outcomes’ are associated with ID comorbidity? Of the many comorbidities that leads to poorer outcomes for people with ASD, why was ID singled out?

2.c. There is insufficient discussion on the effect of familial history.

2.d. p3: '… environmental factors are also likely to play a role': which environmental factors?

2.e. p3: '… during a critical period in pregnancy': more details are needed. Which period specifically is critical for ASD development and how is this relevant under the given context?

2.f. The larger half of the last paragraph (from ‘Information regarding factors associated with pregnancy…’) should be in Methods instead.

3. Methods: This section failed to present the methodologies clearly and led the reviewer uncertain if appropriate methods have been used, most notably regarding samplig, statistical model specification, and variable descrioptions/definitions.

3.a. p5: The Skania Autism Study, the larger project on which the current paper is part of, needs more descriptions e.g. data collection, sample size, demographics, length of follow-up, attrition rate etc.

3.b. p5: Youth- and habilitation center is not a familiar establishment for readers outside Sweden. A brief introduction would be helpful. Also there are mentions of similar terms like ‘habilitation clinic’ elsewhere. If they are one and the same the terminology should be standardised.

3.c. p5: the abbreviation ID (here presumably for identification) is confusing as the same abbreviation has been defined and otherwise used in this manuscript for intellectual disability.

3.d. p6: it is not mentioned if any data derived from the blood samples were used in the current study. If these are not relevant, the blood sample analyses should not be mentioned in the Methods section.

3.e. p7: it is not randomly selected if it is frequency-matched.

3.f. p7: what is the rationale of selecting one (and only one) control for the same type of blood analyses? As discussed, if the blood analyses are irrelevant to the current study this should not be described.

3.g. p8: for a matched case-control study, conditional logistic regression is the appropriate statistical model to account for the clustered structure of the data.

3.h. p8: For most of the variables described here, how the data are collected failed to be sufficiently described. For example, when was maternal maternal smoking and BMI measured? How was involuntary childlessness defined?

4. Results: Here a number of statistical tersm are used in unclear or erroneous manners.

4.a. p15: 'significantly increased ORs' is an incorrect way to describe the results. OR (odds ratio) is already a comparison between (the odds) of two groups.

4.b. p15: ‘Maternal smoking was a significant factor associated with ASD’: this is confusing, is it significantly associated or is it being qualified as significant for other, unspecified, reasons?

4.c. p15 ‘Maternal smoking was a significant factor associated with ASD, for all ASD subgroups’: as far as I can tell from the tables, this is not true for moderate/severe ASD.

4.d. p15: '... A u-shaped association between ASD and BMI was indicated, but far from statistically significant(P>0.9)': It is unclear what statistical test was performed to test the for this u-shaped association. Once again, 'maternal BMI' should be specify to avoid confusion.

5. Discussion

5.a. p24: ‘In our study, information on smoking were obtained during the first trimester’: this information should be in the Methods section.

5.b. p26: Why and how may paternal age be an important confounder? There should be more discussion on this.

5.c. p28: Discussions on strengths and limitations are insufficient. Take limitations as the example, the only one discussed was a lack of access to certain variables. What limitations may have been incurred from the study design, analytical/statistical models, data collection and variable definition, etc?

5.d. p29: More discussion on the policy/clinical/real-world implications is needed.

6. Language: Beyond those mentioned above, a large number of other language and editing problems renders the manuscript difficult to understand and ineffective in conveying the scientific information. It also gives the manuscript the look of an unfinished document. The manuscript needs to undergo extensive and professional English editing if not extensive rewriting. Examples are given in the following pointers but they are not meant to be exhaustive:

6.a. grammar: a comma is missing after ‘compared with ASD with ID’ ( in Introduction, p3); ‘First data from habilitation register and perinatal register accessed 2017-09-25’ is not a complete sentence (Methods, Ethics)

6.b. typo: chronic instead of chronical diseases (Methods, p5)

6.c. clarity: some sentences are difficult to comprehend. For example,’including children to mothers who were born in Sweden only’ (Introduction, p4) is confusing; ‘The medical records from the habilitation clinics were particularly useful…’: in what way useful? specifically how are these medical records used? Or instead, simply ‘The medical records from the habilitation clinics were used to assess level of ASD severity’ if those records were indeed the sole data used for severity triage; overweight and obesity were two terms used in this manuscript for several times but never defined; still another example is the missing ‘maternal’ where it is presumably needed (e,g. Resultsm ‘Furthermore, a strong and significant association between increasing BMI and ASD was found’);

6.d. incorrect word choice: Introduction, p3, (‘Swedish national clinical guidelines assume’) a better verb would be ‘estimate’; Methods p7: strong impression instead of strong hypothesis; Discussion, p26: ‘heterogenic relationships in Sweden’ is confusing. Do you mean heterosexual?

6.e. scientific writing quality: some sentences need to be rewritten to be more informativ, more succinct, or both. For example, in Introduction, p3 (Comorbidity is common and many individuals with ASD face an increased risk of physical and psychological health problems.), the second clause adds little new information to the first. May rewrite it into “Individuals with ASD face an increased risk of physical and psychological health problems, including………”; Discussion, p24: ‘The results are in line with von Ehrenstein et al., who…’: not in line with the researcher, but with their findings or with the paper. In either case the sentence needs rework; Discussion, p25: ‘probably several would follow this advice’ reads conjectural and should be backed by evidence or sounder arguments.

6.f. other English editing issues: unstandardised indentations (e.g. Introduction, paragraph 1 vs the rest); unwarranted shade behind texts (e.g. in Introduction, p5)

Reviewer #2: This study examined maternal and perinatal conditions associated with ASD, integrating population-based registry data with case-level diagnostic validation to enhance reliability. The findings indicate that maternal overweight/obesity is a significant risk factor for ASD, suggesting modifiable maternal health factors that may inform prevention strategies. These findings provide valuable insights into potential ASD risk factors. However, one concern warrants further discussion.

There are several notable findings related to familial ASD, including higher maternal age, elective C-section, and maternal epilepsy, which were associated specifically with familial ASD but not non-familial ASD. To better understand whether these associations are influenced by maternal-specific factors, paternal contributions, or broader familial genetics, further analysis distinguishing maternal vs. paternal familial ASD should be considered. This distinction could help determine whether these associations result from confounding factors, such as parental characteristics or reproductive patterns, or if they reflect genuine interactions between environmental and genetic influences.

**Do you want your identity to be public for this peer review?** For information about this choice, including consent withdrawal, please see our Privacy Policy

Reviewer #1: No

Reviewer #2: No

---

## [Author Response · Author response to Decision Letter 1]

3 Sep 2025

We have added a rebuttal letter that responds to each point raised by the academic editor and reviewers. This letter is labeled "Respons to Reviewers".

---

## [Decision Letter · Decision Letter 1]

30 Oct 2025

Dear Dr. Edlund,

Thank you for submitting your manuscript to PLOS ONE. After careful consideration, we feel that it has merit but does not fully meet PLOS ONE’s publication criteria as it currently stands. Therefore, we invite you to submit a revised version of the manuscript that addresses the points raised during the review process.

We look forward to receiving your revised manuscript.

Kind regards,

Mu-Hong Chen, M.D., Ph.D.

Academic Editor

PLOS ONE

Journal Requirements:

Reviewers' comments:

Reviewer's Responses to Questions

**Comments to the Author**

Reviewer #1: All comments have been addressed

Reviewer #2: All comments have been addressed

Reviewer #3: (No Response)

Reviewer #4: All comments have been addressed

2. Is the manuscript technically sound, and do the data support the conclusions?

Reviewer #1: Yes

Reviewer #2: Yes

Reviewer #3: (No Response)

Reviewer #4: Yes

3. Has the statistical analysis been performed appropriately and rigorously?

Reviewer #1: Yes

Reviewer #2: Yes

Reviewer #3: Yes

Reviewer #4: Yes

4. Have the authors made all data underlying the findings in their manuscript fully available?

Reviewer #1: Yes

Reviewer #2: No

Reviewer #3: (No Response)

Reviewer #4: No

5. Is the manuscript presented in an intelligible fashion and written in standard English?

Reviewer #1: Yes

Reviewer #2: Yes

Reviewer #3: No

Reviewer #4: Yes

Reviewer #1: The authors have satisfactorily addressed the majority of the points raised in the previous comment, either by appropriate revision or cogent responses. The revised manuscript will be a valuable addition to this journal after some relatively minor issues are dealt with.

1. Please standardise the terms related to sex and gender. For example, ‘sex’ was used in the main text and table 2, whereas ‘gender’ was used in table 1. Also, ‘males’ and ‘females’ in table 1 and ‘boys’ and ‘girls’ in table 2. As far as common usage goes, sex is biologically determined and gender is identified/self-reported.

2. Issues regarding familial vs non-familial ASD

2-a. There appears to be no description of how they are defined in the Methods text.

2-b. In Table 2 (and other places), the term heredity was used with the two categories (not reported vs first or second degree). This is confusing, as those with only >2 degree family history are apparently in the ‘not reported (any heredity, presumanly)’ group. It may be clearer to use the term familial vs non-familial consistently throughout with clearly laid-out definitions.

3. In the abstract, it is said that conditional logistic regression (CLR) was used (as recommended in the previous comment), but the main text (and the author’s response) suggested otherwise (they need to be made consistent, firstly). The authors argued that matching (only) on sex and age does not warrant the use of conditional logistic regression. However, any degree of matching creates clusters (matched sets), which violates the independence hypothesis. CLR was used to deal with this. In fact, CLR was commonly used in studies with similarly matching procedure (so called ‘loose-matching’ when N of matched variables is small) (e.g. https://doi.org/10.1016/j.jpsychires.2025.01.028,
https://doi.org/10.1371/journal.pone.0264634, ). On the other hand, there has been methodology papers suggesting that loose-matching data do not always warrant conditional logistic regression (see for example https://doi.org/10.3389/fpubh.2018.00057). It is advised that the authors either:

3-a. Use CLR for the main analysis

3-b. Include CLR as sensitivity analysis

3-c. Discuss (in the Limitation section) the limitation of using (unconditional) logistic regression, and present an evidence-supported argument that loose-matching data does not necessarily need CLR.

Reviewer #2: All my comments have been addressed. I am satisfied with the revision and have no further comments.

Reviewer #3: This is a well-designed, large-scale, population-based case–control study investigating perinatal and maternal factors linked to ASD in a Swedish cohort. The manuscript demonstrates methodological rigor, strong diagnostic verification, and an insightful subgroup analysis stratified by ASD severity, ID status, and familial ASD. The writing is generally clear and logical. However, several issues regarding structure, clarity, and interpretation should be further addressed.

Major Comments

1. The topic is important and timely, but the novelty relative to prior Scandinavian registry studies (e.g., Larsson et al., 2021; Magnusson et al., 2020) should be more explicitly stated.

2. Clarify what this study adds beyond existing large-registry findings—perhaps the strength lies in validated DSM-5 diagnoses and detailed subgroup analyses.

3. While stratification by ASD severity, ID, and familial status is valuable, the rationale for these specific subgroups should be expanded in the Methods or Introduction.

4. Discuss potential power limitations for smaller strata (e.g., severe ASD with ID) and whether multiple-comparison issues were considered.

5. Some findings (e.g., advanced maternal age with familial ASD) could reflect shared heritable traits; this needs a stronger conceptual discussion rather than causal inference.

6. Please clarify whether potential collinearity among maternal variables (e.g., BMI, diabetes, smoking) was examined.

7. The justification for inclusion of variables in multivariable models should be explicit (e.g., all with p < 0.10 in univariate analysis or theory-driven inclusion).

8. Consider sensitivity analyses including paternal age, since maternal and paternal ages are correlated and may confound observed effects.

9. Missing BMI values showed significant associations. The authors should explain the proportion and possible bias (e.g., were missing data more common in certain hospitals or maternal profiles?). Multiple-imputation or complete-case analyses could be compared.

10. The association between elective and emergency C-sections and ASD warrants more nuanced discussion. Distinguish between the effects of indication versus procedure itself; cite recent meta-analyses addressing potential confounding by obstetric complications.

11. Smoking data were collected only in early pregnancy. The limitation that later cessation may bias associations should be clearly acknowledged in the Discussion, and residual confounding by socioeconomic factors or maternal mental health should be considered.

12. The interpretation of these subgroup-specific effects would benefit from mechanistic discussion (e.g., metabolic inflammation, antiepileptic drug exposure).

13. Provide absolute numbers or event counts in supplementary tables to aid in assessing robustness.

14. Tables 3–6 are comprehensive but difficult to read. Consider summarizing key results in forest plots or highlighting significant findings for readability.

15. Ensure consistent presentation of adjusted ORs and confidence intervals, and include total N per subgroup.

Minor Comments

1. Abstract

(1) The results paragraph could quantify the main associations (e.g., “Maternal obesity [aOR 1.76, 95% CI 1.46–2.12] and smoking [aOR 1.49, 95% CI 1.22–1.82] were linked to increased ASD risk”).

(2) Consider shortening the background to make space for key quantitative findings.

2. Terminology

(1)Use consistent phrasing: “maternal epilepsy” vs. “maternal epilepsia”; “perinatal complications” vs. “pregnancy/delivery complications.”

3. Formatting

(1) Ensure figures and tables follow PLOS ONE format (two-decimal ORs, uniform column alignment).

(2) Check for minor typographical errors (e.g., spacing in “1.28 1.39” in Abstract).

4. Limitations

(1)Add a paragraph explicitly listing main limitations: lack of paternal data, residual confounding by socioeconomic factors, and self-reported smoking/BMI data.

5. Ethical Statement

(1)Clarify whether data linkage was conducted under GDPR-compliant procedures and specify data storage duration.

Reviewer #4: The author has revised this version in accordance with the reviewers’ previous comments. Therefore, no further revisions are required, and the manuscript can be accepted for publication.

**Do you want your identity to be public for this peer review?** For information about this choice, including consent withdrawal, please see our Privacy Policy

Reviewer #1: No

Reviewer #2: No

Reviewer #3: No

Reviewer #4: No

---

## [Author Response · Author response to Decision Letter 2]

8 Jan 2026

Dear Editor,

We thank you and the reviewers for the careful and constructive evaluation of our manuscript. We have revised the manuscript in response to all comments raised during the review process. Below, we provide a detailed, point-by-point response describing how each comment has been addressed in the revised version.

Editorial comment on wording and phrasing

As specifically requested by the Academic Editor, Dr. Mu-Hong Chen, we carefully reviewed the wording and phrasing throughout the entire manuscript to ensure that the language is appropriate and not potentially harmful or offensive to the autistic community. In particular, we addressed concerns regarding wording that could be interpreted as implying prevention of autism.

In the revised manuscript, we clarified that terms such as “risk” and “association” are used strictly in an epidemiological, non-normative sense, consistent with the observational design of the study. Specifically, we revised wording that could be construed as prevention-oriented and replaced it with language emphasizing etiological understanding, subgroup heterogeneity, and how maternal and perinatal health contexts relate to neurodevelopmental diversity. The revised text is framed within a neurodiversity-informed and epidemiological perspective and does not suggest preventive goals.

Responses to Reviewers

Reviewer #1

We thank Reviewer #1 for the overall positive assessment of the revised manuscript and for the helpful suggestions.

Comment 1: Standardize terminology related to sex and gender

Response:

We have revised the entire manuscript, including all tables and figure captions, to ensure consistent terminology. We now use sex throughout when referring to biological classification and consistently refer to males and females rather than age-specific terms such as boys and girls. References to gender have been removed to avoid ambiguity.

Comment 2: Definition and terminology of familial vs. non-familial ASD

Response:

We have clarified and made explicit the definition of familial versus non-familial ASD in the Methods section. Specifically, cases are now defined as familial ASD if at least one first- or second-degree relative had a documented ASD diagnosis, while non-familial ASD refers to the absence of any reported ASD diagnosis among close relatives rather than missing information.

In addition, terminology has been harmonized throughout the manuscript and tables by consistently using the terms family history, familial ASD and non-familial ASD, replacing the previously used term heredity.

Comment 3: Use of conditional vs. unconditional logistic regression

Response:

We thank the reviewer for raising this important methodological point. We maintain that unconditional logistic regression was the most appropriate approach for the present study design. A conditional logistic regression model can be justified when cases and controls are closely matched on multiple background variables, apart from outcome status. In our study, such extensive matching was not performed, as it would have precluded evaluation of several exposures of interest. Instead, we applied a loose-matching design, frequency-matching cases and controls on sex and year of birth only. Accordingly, we follow the reviewer’s alternative 3c. We have added a paragraph in the Discussion addressing this methodological choice and citing relevant methodological literature on the performance of conditional versus unconditional logistic regression in loosely matched data. In addition, we have clarified the matching strategy and justification for unconditional logistic regression in the Methods section, supported by methodological evidence indicating that this approach yields valid estimates when matching variables and relevant covariates are included (Kuo et al., 2018). Finally, we corrected an erroneous statement in the abstract regarding the use of conditional logistic regression, which arose due to a misunderstanding between authors.

Reviewer #2

Response:

We thank Reviewer #2 for the positive evaluation and for confirming that all comments have been addressed.

Reviewer #3

We thank Reviewer #3 for the detailed and constructive comments.

Comment 1–2: Novelty relative to prior Scandinavian registry studies

Response:

We agree that the novelty of our study required clearer articulation. We have therefore added a new section entitled “Novelty and Contribution” immediately before the Strengths and Limitations section. This section clarifies how our study extends prior Nordic register research by using individually confirmed DSM-5 diagnoses, restricting analyses to mothers born in Sweden to reduce heterogeneity related to migration and ethnicity, and examining subgroup-specific association patterns by ASD severity, intellectual disability, and familial ASD. A brief reference to a recent large Nordic register study has also been added to the Discussion to contextualize our findings.

Comment 3: Rationale for subgroup stratification

Response:

We added a short explanatory paragraph in the Introduction clarifying the rationale for stratifying by ASD severity, intellectual disability, and familial ASD. The text explains that these dimensions capture both clinical and etiological heterogeneity, as severity and intellectual disability have been linked to partly distinct neurodevelopmental and genetic mechanisms, while familial ASD reflects inherited liability that may modify environmental associations.

Comment 4: Power limitations and multiple comparisons

Response:

We have revised the Strengths and Limitations section to explicitly address potential power limitations in smaller subgroup strata, particularly severe ASD and ASD with co-occurring intellectual disability, where sample sizes were limited and confidence intervals wider. We now clarify that subgroup-specific estimates should therefore be interpreted with caution. We also explicitly acknowledge that the large number of subgroup analyses increases the risk of chance findings due to multiple comparisons. Accordingly, we emphasize interpretation based on effect sizes, confidence intervals, and consistency of patterns across related analyses rather than on statistical significance alone.

Comment 5: Advanced maternal age and heritable traits

Response:

We expanded the conceptual discussion in the Maternal Age section to clarify that associations between advanced maternal age and ASD, particularly in the familial subgroup, may reflect shared heritable traits and assortative mating rather than intrauterine causal mechanisms. Relevant references were added to support this interpretation.

Comment 6: Collinearity among maternal variables

Response:

Maternal BMI, smoking, and diabetes were included simultaneously in the multivariable models. While we did not perform a formal collinearity diagnostics, we did not observe indications of model instability or implausible coefficient estimates when these variables were entered together. We therefore consider the risk of problematic collinearity to be limited, but acknowledge this as a potential methodological consideration.

Comment 7: Justification for variable inclusion

Response:

Variables were included based on prior epidemiological evidence and data availability rather than statistical selection procedures. Specifically, included factors were those that have previously been associated with ASD or with pregnancy and delivery complications, as now explicitly stated in the Statistical Analyses section.

Comment 8: Paternal age

Response:

We agree that paternal age may be an important confounder, given its correlation with maternal age and its established association with ASD risk. Unfortunately, information on paternal age was not available and could therefore not be included in sensitivity analyses. As discussed in the manuscript, maternal and paternal ages are strongly correlated in Sweden, and observed associations with maternal age likely reflect shared parental age-related and familial factors. This limitation is explicitly acknowledged.

Comment 9: Missing BMI data

Response:

Missing BMI values were present in 8.3% of ASD cases and 5.8% of controls, suggesting that missingness was not random. Rather than applying imputation methods that assume data are missing at random, we therefore treated missing BMI as a separate category. As discussed in the Discussion, we suspect that non-response may be related to underlying maternal characteristics, including overweight or obesity. To assess robustness, we also performed complete-case analyses; estimates for the other maternal covariates changed only marginally compared with the main analyses, indicating that the overall findings were not driven by the handling of missing BMI data. We note that the study was conducted within a single regional public healthcare system with standardized antenatal care routines, making systematic hospital-level differences in BMI recording unlikely.

Comment 10: Cesarean delivery

Response:

We added a statement in the Discussion noting that associations between cesarean delivery and ASD may reflect confounding by indication rather than procedure-specific effects, supported by large meta-analyses and sibling-based studies.

Comment 11: Smoking data

Response:

We clarified that smoking was assessed only at the first antenatal visit and that later changes in smoking behavior could not be captured. This limitation is now explicitly discussed, along with the potential for residual confounding by socioeconomic and maternal mental health factors.

Comment 12: Mechanistic discussion

Response:

A brief statement was added noting that future studies integrating biological data may help clarify underlying mechanisms.

Comments 13–15: Tables and figures

Response:

Supplementary tables were expanded to include absolute numbers and total subgroup sizes. Key findings are now summarized using forest plots to improve readability, and presentation of adjusted odds ratios and confidence intervals has been standardized.

Minor comments and formatting

Minor typographical issues were corrected, terminology harmonized, limitations expanded, and the Ethical Considerations section updated to clarify GDPR-compliant data handling and storage procedures.

Reviewer #4

Response:

We thank Reviewer #4 for the positive evaluation and are pleased that the manuscript is now considered acceptable for publication.

---

## [Decision Letter · Decision Letter 2]

2 Feb 2026

Perinatal and maternal factors associated with Autism Spectrum Disorder.

PONE-D-24-58795R2

Dear Dr. Susanna Edlund,

We’re pleased to inform you that your manuscript has been judged scientifically suitable for publication and will be formally accepted for publication once it meets all outstanding technical requirements.

Kind regards,

Mu-Hong Chen, M.D., Ph.D.

Academic Editor

PLOS One

Additional Editor Comments (optional):

Reviewers' comments:

Reviewer's Responses to Questions

**Comments to the Author**

Reviewer #3: All comments have been addressed

2. Is the manuscript technically sound, and do the data support the conclusions?

Reviewer #3: Yes

3. Has the statistical analysis been performed appropriately and rigorously?

Reviewer #3: Yes

4. Have the authors made all data underlying the findings in their manuscript fully available?

Reviewer #3: Yes

5. Is the manuscript presented in an intelligible fashion and written in standard English?

Reviewer #3: No

Reviewer #3: Overall, the authors have made substantial efforts to address the major concerns raised in the previous review rounds. Importantly, they have now clearly acknowledged the key limitations of the study, including the small sample size, the absence of an ADHD-only comparison group, and the potential influence of unmeasured confounding factors. The statistical analyses have been strengthened through the use of appropriate corrections and effect size reporting, and the manuscript is now more transparent in its methodological description. While the study remains exploratory in nature, the authors have appropriately tempered their interpretations and framed the findings as preliminary.

**Do you want your identity to be public for this peer review?** For information about this choice, including consent withdrawal, please see our Privacy Policy

Reviewer #3: No

---

## [Editor Report · Acceptance letter]

PONE-D-24-58795R2

PLOS One

Dear Dr. Edlund,

I'm pleased to inform you that your manuscript has been deemed suitable for publication in PLOS One. Congratulations! Your manuscript is now being handed over to our production team.

Kind regards,

on behalf of

Dr. Mu-Hong Chen

Academic Editor

PLOS One